# Cost Efficiency and CO$_2$ Emission Reduction in Short Sea Shipping: Evidence from Ciwandan Port–Panjang Port Routes, Indonesia

Dedy Arianto [1], Edward Marpaung [1], Johny Malisan [1], Windra Priatna Humang [1,*],
Feronika Sekar Puriningsih [1], Mutharuddin [1], Tetty Sulastry Mardiana [1], Wilmar Jonris Siahaan [2],
Teguh Pairunan [2] and Abdy Kurniawan [2]

[1] National Research and Innovation Agency, Jakarta Pusat 10340, Indonesia; ariantodedy17@gmail.com (D.A.);
edpaung@yahoo.com (E.M.); joylisann@gmail.com (J.M.); ferospuriningsekar@gmail.com (F.S.P.);
mutharuddin@gmail.com (M.); tetty_sulastry@yahoo.com (T.S.M.)

[2] Transport Policy Agency, Ministry of Transportation, Jakarta Pusat 10110, Indonesia;
siahaan.wilmar12@gmail.com (W.J.S.); teguhputra661@yahoo.com (T.P.); birulaut09@hotmail.com (A.K.)

* Correspondence: windra.priatna.humang@brin.go.id

**Abstract:** Merak Port of Java and Bakauheni Port of Sumatera are connected by ferry lines. However, the number of ferry ships and facilities of the two ports are not able to accommodate the number of vehicles that cross. Queues of vehicles often occur and waiting times at the port are very high and have an impact on the accumulation of vehicles on the road to the port. Anticipating these conditions, it is possible to open a short sea shipping (SSS) route from Ciwandan port to Panjang port as an alternative route for shifting some of the vehicles served by those ferry ships. This research aims to analyze the efficiency of opening the Ciwandan–Panjang SSS route in terms of benefits for stakeholders, cost efficiency for vehicle users, and the potential for CO$_2$ reduction from exhaust gases. We use a descriptive quantitative method. The analytical techniques used include port cost analysis, benefit analysis (for the government and ship operators), comparative analysis of transportation costs, and analysis of the impact of reducing CO$_2$ emissions, which are valued monetarily. The results of the analysis show that the operation of the Ciwandan–Panjang SSS can reduce the total cost of vehicles compared to the Merak–Bakuheni route. Owners of cargo vehicles are able to save on logistics costs of IDR 332 billion per year. Estimated state revenue through non-tax state revenues (NTSI) and value-added tax (VAT) is increased. Losses due to CO$_2$ emissions are estimated to be reduced, with a value of up to IDR 511 billion per year.

**Keywords:** short sea shipping; environmental sustainability; emission; cost efficiency

## 1. Introduction

Sea transportation is an important component of transportation networks around the world. Sea transportation, which includes deep-sea shipping and short sea shipping (SSS), has long been a major facilitator of economic progress and prosperity in Europe [1]. SSS is the delivery of commodities by sea over a relatively short distance, in contrast to deep sea shipping, which is transcontinental and takes place on oceans [1–3]. In fact, SSS is a viable alternative to road transport [4], because of its role in reducing traffic congestion and emissions from long-distance cargo transportation and environmental concerns, respectively [5,6]. The European Commission's white paper [7] on transport suggests that by 2050, EU CO$_2$ emissions from sea transport should be reduced by at least 40% from 2005 and, if possible, by 50%.

SSS started in 1982 and was initiated by Balduini, who said that SSS was maritime transportation between ports of a nation as well as between ports of a country and ports of a bordering country. In 1993, Criley and Dean defined SSS based on the maximum size

of the ship, up to 5000 gross tonnages. The same was stated by Bagchus and Kuipers [8], who defined SSS based on the size of the ship. Bjornland [9] stated that SSS was the transportation of goods transported by sea without crossing the ocean. SSS has been described based on technical criteria of ship size and type, cargo transported (cargo), port, network, and information system [10,11]. Stopford [12] saw SSS as a feeder service in competition with road services and allowed for modal shifts in freight transport. According to Douet Marie [13] on the Committee of the European Union, SSS could be defined as "Maritime Highway Transport Systems" and included canals, rivers, and other inland waterways, as well as coastal shipping systems. Transport from or to inland rivers was also considered SSS [14].

SSS is the maritime transport of goods over relatively short distances, as opposed to intercontinental cross-ocean transport. In the context of European Union (EU) transport statistics, it is defined as the sea transport of goods between ports in the EU (including candidate countries and EFTA countries), the Mediterranean, and the Black Sea.

In global areas such as the United States of America, the application of coastal ferries or marine highways to several routes (California–Washington, Florida–Maine) is affected by: population growth; increased fuel consumption and reduced productivity as a result of congestion; and damage to the highway due to overloading. The use of SSS in various parts of the world has been shown to reduce mileage, congestion and road loads, fuel consumption, and $CO_2$ emissions, all of which are expected to improve logistics operational efficiency [15–20].

SSS is considered an economical, sustainable, and competitive mode of transportation [11], reducing traffic congestion and increasing economic development [21]. In addition, according to Jurcovic et al. [21], SSS also provides benefits to industrial sectors with an average cost savings of more than EUR 32,000 per km. The cost of developing SSS infrastructure (12,600 EUR per km) is lower than that of road transport (45,210 EUR per km) [7].

Several studies comparing the effectiveness of SSS with other modes of transportation showed a significant reduction in costs. Suarez-Aleman et al. [22] obtained competitive costs from implementing SSS with Spanish routes to several destinations in Europe (London, Paris, Rome, Berlin, and Moscow), taking into account monetary costs, time costs, and external costs. Martinez-Moya and Feo-Valero [23] developed the Port Connectivity Index and applied it to Spanish ports to study SSS connectivity in container transport. Santos, T. A. et al. [24] found cost efficiency using intermodal transport (SSS–inland waterway, SSS–train) in the Atlantic corridor with numerical analysis. Research by Kim N.S. et al. [25] found that SSS-based intermodal transportation is cheaper than road-based transportation, especially over 2000 km distances. They also found that, according to user preferences, the time factor seems to be more important than transportation costs. In addition, SSS transportation also offers higher fuel economy and lower emissions of harmful pollutants. To integrate SSS into multimodal transport networks, regulators should develop policies that take into account the needs of shippers and stimulate innovation within the industry, as innovation can minimize unnecessary costs (e.g., port fees) [26,27].

Problems that occur in European countries also occur in Indonesia, especially for the movement of goods from Java to Sumatra. The flow of vehicles is still dominated by land modes and the Merak–Bakauheni ferry crossing, so sea transportation has not been utilized properly. The impact is a decrease in the quality of land transportation infrastructure services, such as traffic congestion, high accident rates, pollution, energy wastage, and road damage due to trucks exceeding the load limit [28]. This results in high transportation costs, coupled with external costs of land transportation [29]. Currently, the dominant mode of transportation is for the distribution of logistics goods from Java to Sumatra, or vice versa, using road transportation and the Merak–Bakauheni crossing. According to the results of research conducted by the Indonesian Trucking Association [30], on the island of Sumatra, the land transportation mode dominates, at 95.20%, while the sea transportation mode is only 2.70%. Meanwhile, in Java, the share of road transportation is 99.70%, while

sea transportation is 0.20%. The burden on road transportation modes is already very high, while sea transportation has played little role [31,32].

The tendency to use road transportation for logistics transportation in the Java–Sumatra corridor based on operator preferences is, among others, caused by the ability to provide door-to-door services, high accessibility and flexibility, fuel costs that still receive subsidies from the government, the high availability of transportation equipment at any time, and short delivery times, and load limit violations are still possible. However, in the long run, this will result in large costs incurred due to road repairs reaching 49,277 EUR per km and a large environmental impact caused by vehicle exhaust gases, which have a negative impact on health [33].

The difference between this research and previous research lies in the indicators analyzed. This study explains the possibility of implementing SSS for the Ciwandan Port–Port of Panjang route by analyzing cost efficiency based on the benefits obtained from the vehicle operator side, government revenue, and the impact of $CO_2$ reduction from vehicle exhaust gas, which is valued in terms of money. The implication of this research is the government's policy towards diverting logistical vehicle routes from ferry transport to seaports.

In addition to this introduction, this paper consists of three other sections. The second section presents the methodology, which includes the research location, research approach, and analytical techniques used. The third section presents the results of the analysis, consisting of the following subsections: (a) Merak–Bakauheni ferry route service condition; (b) Ciwandan–Panjang route service condition; (c) port cost feasibility analysis; (d) the benefits of diverting the flow of vehicles to the Ciwandan–Panjang route; (e) cost efficiency analysis; (f) estimated reduction in $CO_2$ emissions; and (g) the operational impact of the Ciwandan–Panjang SSS. The fourth concluding section consists of a summary of results, limitations, and suggestions for future research.

## 2. Methods

The methodology for analyzing cost efficiency and $CO_2$ reduction from SSS can be seen in the flow chart Figure 1.

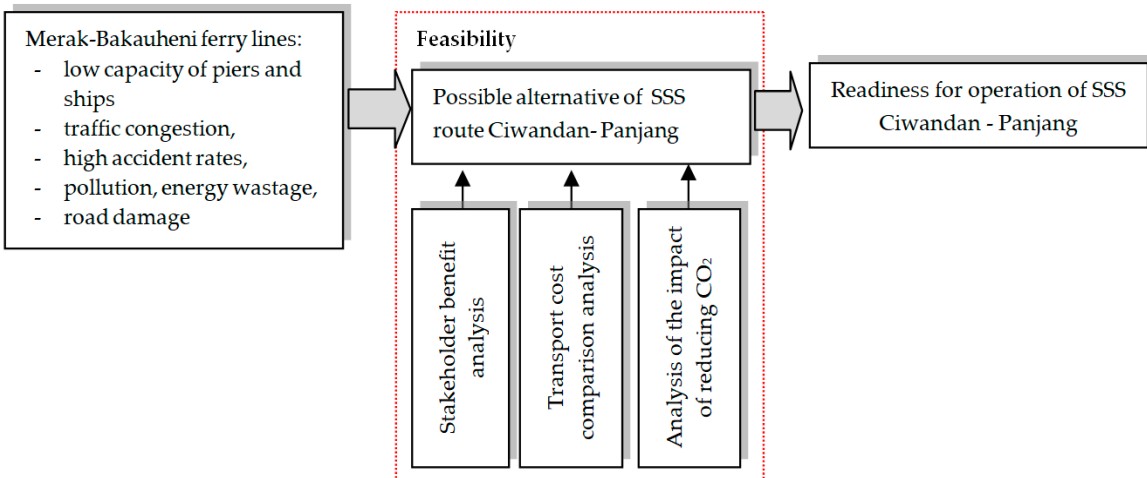

**Figure 1.** Research Method Flowchart.

The research approach used is a combination of qualitative and quantitative. Interviews were conducted with the relevant stakeholders, including vehicle owners, port operators (Indonesian Port Corporation and PT ASDP Indonesia Ferry), and the government. Data was collected through in-depth interviews and questionnaires, as well as focus group discussions (FGD).

The data analysis technique used includes the analysis of port costs, cost benefits (for the government and ship operators), comparative analysis of transportation costs, and analysis of the impact of reducing $CO_2$ emissions due to route changes.

Based on a study by the Indonesian Trucking Association [30], most of the current vehicle movement from Java to Sumatra moves to the city of Bandar Lampung and other cities in Sumatra. Therefore, the research was carried out on the Merak–Bakuheni Ferry route and the Ciwandan–Panjang port route, which include Banten Province and Lampung Province, as shown in Figure 2.

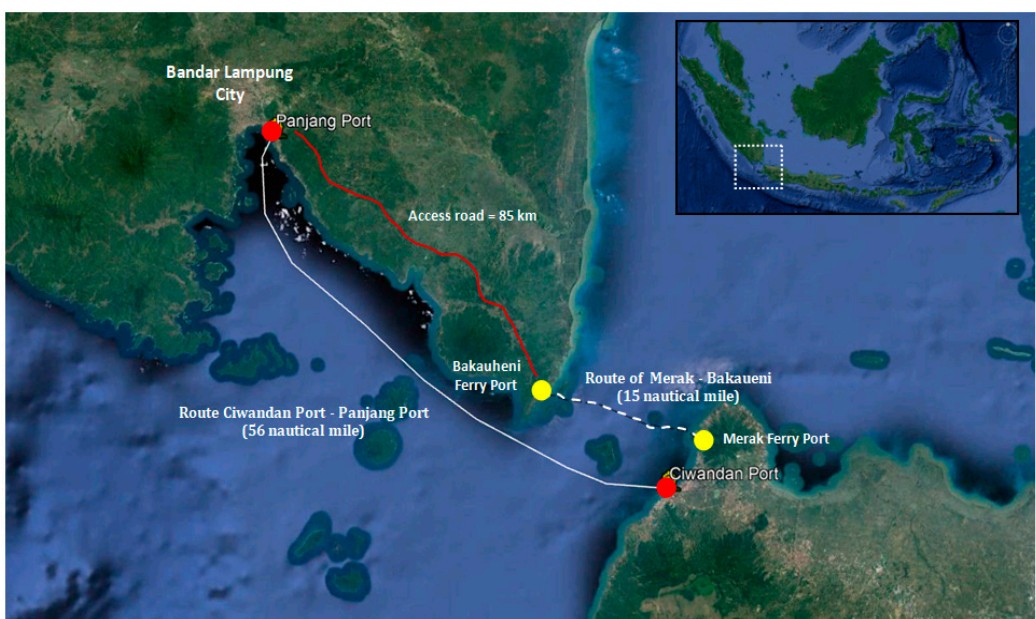

**Figure 2.** Research location (Ferry Merak–Bakauheni and alternative SSS of Ciwandan–Panjang).

*2.1. Port Cost Analysis*

The calculation of port services use the sample ship, KM Mutiara Berkah I, capacity 32,645 GT, with the formulation shown in Table 1.

**Table 1.** Port fee formulation.

| Formulation | Fees at the Port of Panjang | Fees at the Port of Ciwandan * |
|---|---|---|
| Anchorage cost | 85 × GT | 87 × GT |
| Navigation cost | 250 × GT | 250 × GT |
| VTS cost | 200,000 | 200,000 |
| Quarantine cost | 350,000 | 350,000 |
| Berth cost | Etmal (0.5) × tariff (102) × GT | Etmal (0.5) × tariff (68) × GT |
| Pilotage cost | Maneuvering × (tariff fixed + (tariff var × GT)) | |
| Towage cost | Hours of use × (tariff fixed + (tariff var × GT)) | |
| Pass cost | 2000/day = 2000/12 trip × tariff | |
| Freshwater cost | 20 tonnes/trip | 20 tonnes/trip |

Note: * fees at Ciwandan port, use fees at Merak port.

*2.2. Benefit Analysis of Stakeholders*

Benefits obtained by stakeholders (government, port operators, ship operators, and vehicle owners) are one of the reasons for the possible shift in the flow of vehicle movement from the Merak–Bakuheni route to the Ciwandan–Panjang route. Therefore, it must be ensured that stakeholders will benefit when there is a vehicle transition.

Benefits will be obtained by the government through non-tax state income (NTSI) services as well as from value-added tax (VAT). Benefits will be obtained by shipping operators in the form of an increase in the ship's load factor due to the transition of vehicle

loads. Benefits are obtained by vehicle owners through reductions in transportation costs, by comparison between the Merak–Bakuheni and Ciwandan–Panjang routes.

### 2.3. Impact Analysis of $CO_2$ Emissions

The reduction in the impact of $CO_2$ emissions is calculated using the IPCC emission factor formulation [34], based on the probability of vehicles switching from Merak–Bakuheni to Ciwandan–Panjang.

## 3. Results

### 3.1. Merak–Bakuheni Ferry Route Service Conditions

The Merak–Bakuheni route is currently served by 70 ships of different sizes and capacities, dominated by ships with a size of 5000–10,000 GT and a ship age of 26–35 years (Figure 3). Vessel utilization is very low due to the large number of ships operating on the Merak–Bakuheni route, which is not proportional to the capacity of the port dock. Most ship operators obtain a low income, not proportional to the costs, because the operating times of ships are very low. This condition causes ships with a size of 5000 GT to be moved to other routes outside Java.

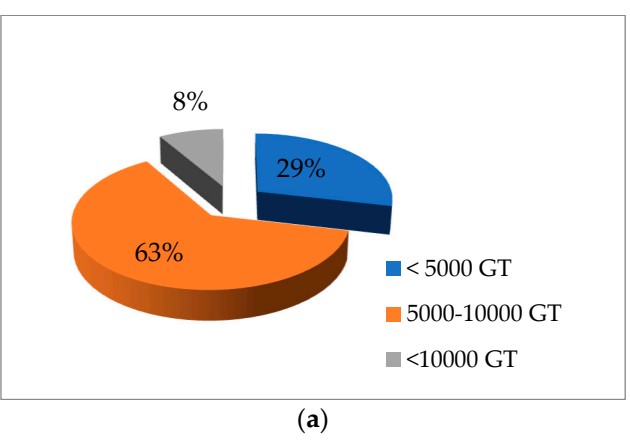

(**a**)

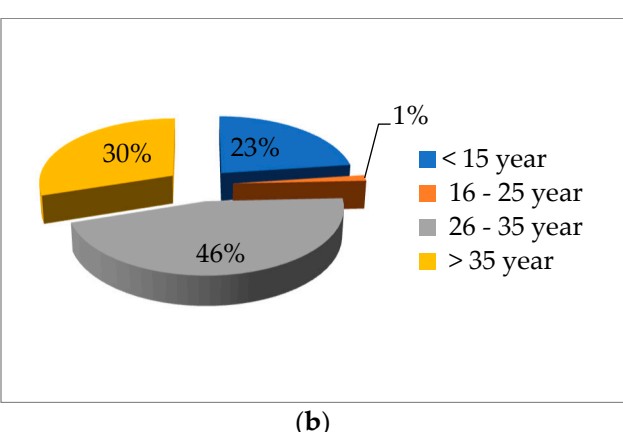

(**b**)

**Figure 3.** Characteristics of the size (**a**) and age of the ships (**b**) on the Merak–Bahauheni route.

The operating pattern of ships served by six docks is a sailing time of 120 min (sailing time, time of entry and exit), a port time of 60 min (unloading and loading, service time for passenger and vehicle loads, and ticket claim time), manifest printing, as well as ship clearance processing time, a cargo service time of 12 min (time spent directing passenger and vehicle loads), and a ramp door closing time of 20 min.

The number of vehicles that pass through the Merak–Bakuheni route fluctuates every day. The results of the vehicle traffic survey at the port in August 2018 will be the basis for calculating the average vehicles per day, as shown in Figure 4.

The average number of vehicles that cross the Merak–Bakuheni is 5285 per day. Freight transport dominated by up to 52%, private transport 43%, and public transport 5% (Figure 5a). When compared based on "peak season" conditions, there is a very high spike. The comparison of the number of vehicles per ship trip based on peak season and non-peak season conditions is shown in Figure 5b.

Based on surveys and interviews with vehicle owners that cross the Merak–Bakuheni route, as many as 36% (2000 vehicles) have the potential to switch to using the Ciwandan–Panjang SSS. This amount is also assumed to be able to be accommodated by the Ciwandan–Panjang port according to the current pier capacity.

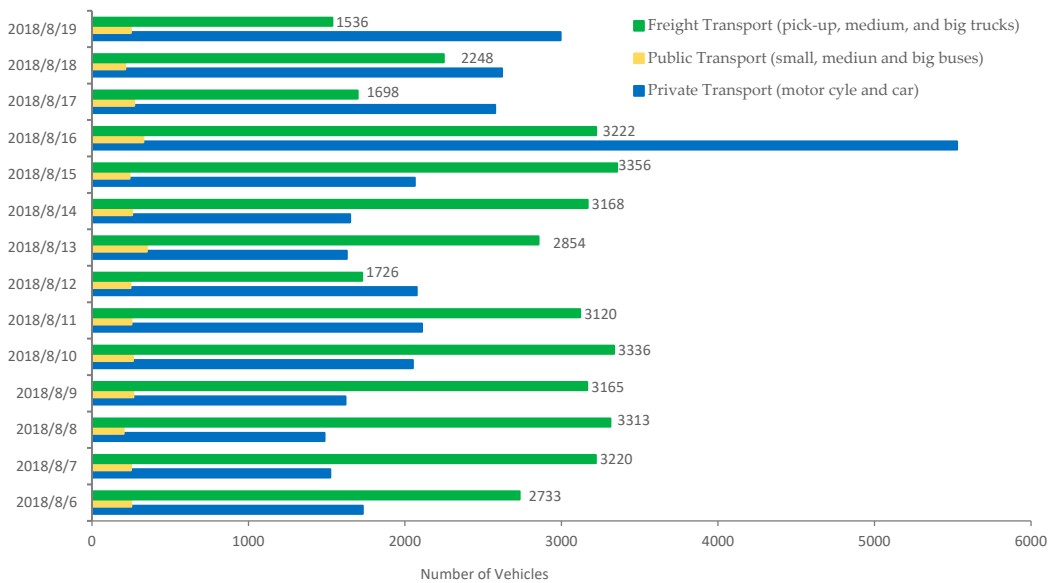

**Figure 4.** Average vehicles crossing per day.

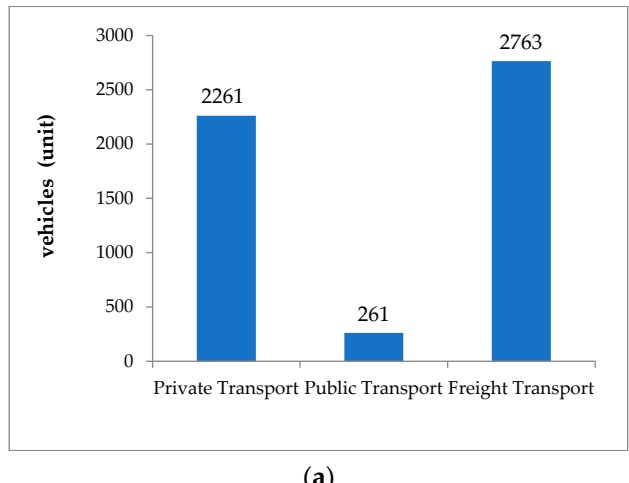

(**a**)

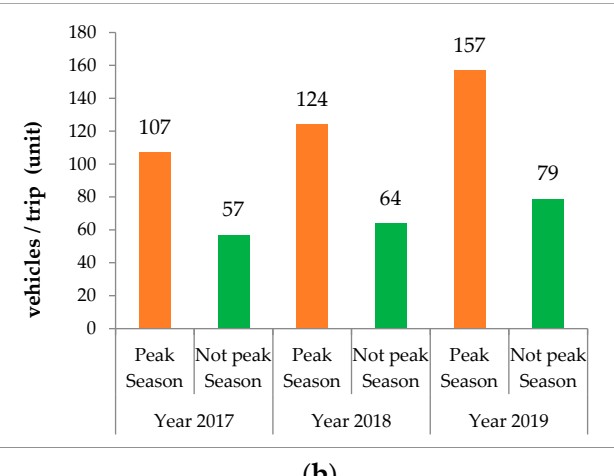

(**b**)

**Figure 5.** (**a**) Average number of vehicles crossing Merak–Bakuheni by type; (**b**) average vehicle per trip during peak and non-peak seasons.

### 3.2. Ciwandan–Panjang Route Service Conditions

Ciwandan Port, which is located in Banten Province, and the Port of Panjang, which is located in Lampung Province, are seaports that currently operate to serve domestic and foreign transportation. Ship visits to these two ports fluctuate every year (Figure 6).

At Ciwandan port, there are seven docks with differing capacities. However, according to the data, the berth occupancy ratio (BOR) is still low. The BOR at Ciwandan port is 45.47%, while at Panjang port it is 51.2%. Therefore, dock services at the two ports can still be improved by adding more ship visits for SSS services. With these BOR conditions, it is assumed that there are still two docks that can be optimized to serve the Ciwandan–Panjang SSS.

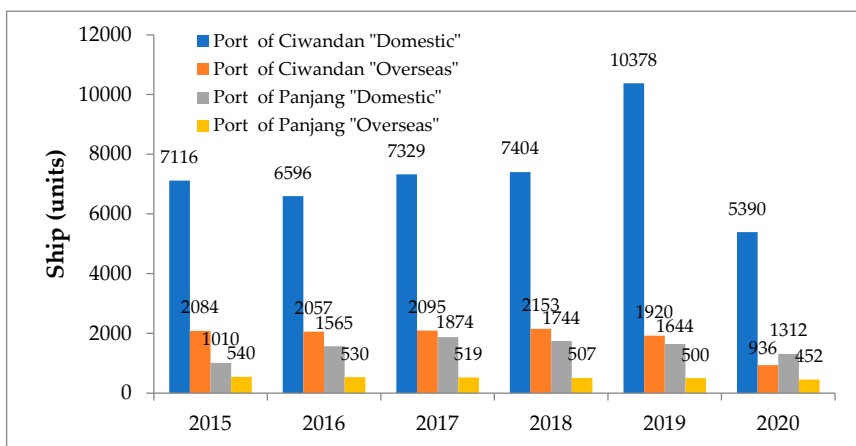

**Figure 6.** Ship visits at Ciwandan and Panjang ports.

*3.3. Port Cost Feasibility for the SSS Ciwandan–Panjang Route*

SSS is operated under the following assumptions: the number of vehicles is 2000 units; a frequency of three round trips; a sailing time of 4 h; a shipload capacity of 167 trucks; a cargo capacity of 30–60 tons/truck; the fleet operated is comprised of 5–6 units. The basis for calculating the port service rates is as shown in Table 2.

**Table 2.** Revenue from port services of the SSS Ciwandan–Panjang.

| Port Service Cost | Calculation | Revenue per Round Trip (IDR) |
|---|---|---|
| Anchorage cost | | |
| -     Panjang port | 87 × 32,645 | 2,840,115 |
| -     Ciwandan port | 87 × 32,645 | 2,840,115 |
| Navigation cost | | |
| -     Panjang port | 250 × 32,645 | 8,161,250 |
| -     Ciwandan port | 250 × 32,645 | 8,161,250 |
| VTS cost | | |
| -     Panjang port | 200,000 | 200,000 |
| -     Ciwandan port | 200,000 | 200,000 |
| Quarantine cost | | |
| -     Panjang port | 350,000 | 350,000 |
| -     Ciwandan port | 300,000 | 350,000 |
| Berth cost | | |
| -     Panjang port | | |
| Masa I | 0.50 × 102 × 32,645 | 1,664,895 |
| Ramp door 25% × time I | | 416,224 |
| -     Ciwandan port | | |
| Masa I | 0.50 × 68 × 32,645 | 1,664,895 |
| Ramp door 25% × time I | | 416,224 |
| Pilotage cost | | |
| -     Panjang port | 2 × (329,472 + (55 × 32,645)) | 5,099,873 |
| -     Ciwandan port | 2 × (217,140 + (62.04 × 32,645)) | 5,099,873 |
| Towage cost | | |
| -     Panjang port | 2 × (4,216,000 + (8 × 32,645)) | 8,954,320 |
| -     Ciwandan port | 4.5 × (2,310,000 + (4.52 × 32,645)) | 8,954,320 |
| Pass cost | | |
| -     Panjang port | 167 × 200,000 | 33,400,000 |
| -     Ciwandan port | 167 × 200,000 | 33,400,000 |
| Freshwater cost | 20 tonnes/trip | |
| -     Panjang port | 20 × 40,600 | 812,000 |
| -     Ciwandan port | 20 × 142,006 | 2,840,115 |

Based on the results of the revenue analysis of port fees per round trip (Table 3), the largest revenue was received from vehicle ticketing costs, towage costs, and navigation costs, while the lowest was from VTS costs. If we compare the income received from port services between the Ciwandan port and the Panjang port, the difference is not significant, considering that the cost of each port is the same as determined by the Indonesian Port Corporation (IPC).

**Table 3.** The benefits that will be obtained by the government and the port operator.

| Port Services Cost | Panjang Port | Ciwandan Port | Government NTSI (IDR) | IPC/Port (IDR) | VAT (IDR) |
|---|---|---|---|---|---|
| Anchorage cost | 2,840,115 | 2,840,115 | 5,680,230 | | |
| Navigation cost | 8,161,250 | 8,161,250 | 362,722 | | |
| VTS cost | 200,000 | 200,000 | 400,000 | | |
| Quarantine cost | 350,000 | 350,000 | 700,000 | | |
| Berth cost | 2,081,119 | 2081,119 | 104,056 | 4,058,181 | |
| Pilotage cost | 5,099,873 | 5,099,873 | 509,987 | 9,689,758 | |
| Towage cost | 8,954,320 | 8,954,320 | 895,432 | 1,7013,208 | |
| Pass cost | 33,400,000 | 33,400,000 | 1,670,000 | 65,130,000 | 6680,000 |
| Freshwater cost | 812,000 | 2,840,115 | | 1,624,000 | |
| Round trip | | | 10,322,427 | 97,515,148 | 6,680,000 |
| Per trip | | | 5,161,214 | 48,757,574 | 3,340,000 |
| Per day | | | 61,934,565 | 585,090,887 | 40,080,000 |
| Per month | | | 1,858,036,939 | 17,552,726,619 | 1,202,400,000 |
| Per year | | | 22,296,442,270 | 210,632,719,426 | 14,428,800,000 |

Description: Anchorage cost, navigation cost, VTS cost, quarantine cost (100% is NTSI), berth cost (2.5% is NTSI), pilotage and towage cost (5% is NTSI), and navigation cost are paid once a month, so the average trip is = IDR 181,361 or IDR 362,722 per round trip.

### 3.4. Benefits of Diverting the Flow of Vehicles to the Ciwandan–Panjang Route

SSS operations at Ciwandan Port and Panjang Port are under the authority of the Indonesian Port Corporation (IPC), a state-owned company that operates the port. Any income earned from ship service fees at the port will be calculated as income to the state in the form of non-tax state income (NTSI) plus value-added tax (VAT), and corporate income in the form of IPC income. The calculation of the value of benefits that will be obtained by the government through NTSI, VAT, and IPC as port operator, is as shown in Table 4.

Based on the analysis of Table 4, the revenue received by the government from NTSI + VAT is only 14.8%, and the operator (IPC) receives 85.2% of the total revenue. In one round trip, the government's revenue reaches IDR 10.3 million, while that of the IPC is IDR 97.5 million. In one day, the government's income reaches IDR 61.9 million, while that of the IPC is IDR 585 million. In one month, the government's income reaches IDR 1858 million, while the IPC's is IDR 17,552 million. In one year, the government's income reaches IDR 22,296 million, while the IPC's is IDR 210,632 million.

**Table 4.** Shipping company benefits.

| Description | Unit | Total |
|---|---|---|
| Number of trucks crossing Java Island–Sumatra Island | truck | 2000 |
| Ship capacity (32,645 GT) | truck | 186 |
| Number of trips/ship/day | trip | 3 |
| Number of trips needed to move 2000 trucks | trip | 11–12 |
| BEP is at *Load Factor* | % | 67.5 |
| Number of ships owned | ships | 4–6 |
| The average load to move 2000 trucks in 12 trips | unit | 167 |
| The load factor is 167/186 | % | 90.0 |

In addition to the benefits obtained by the government and port operators (IPC), shipping companies (shipping operators) also obtain benefits. The benefits obtained by ship operators are the number of vehicles that cause the ship's load factor level to be high. Assuming that it takes 4–6 ships with a size of 32,645 GT (186 vehicles capacity), then to serve 2000 vehicles per day, three trips are needed with an average load factor of 90%. When compared to the operational break even point (BEP) value of the ship, a load factor of only 67.5% is necessary. This shows that the shipping company will experience an advantage when serving the SSS Ciwandan–Panjang route. The calculation of the value of benefits that will be obtained by the shipping company is as shown in Table 4.

Based on the calculations in Table 4, the implementation of the Ciwandan–Panjang SSS as an alternative to Merak–Bakuheni will be able to attract ship operators to invest on the assumption that the estimated load factor is higher than the BEP load factor.

### 3.5. SSS Cost Efficiency Analysis

To assess the cost efficiency of implementing SSS for the Ciwandan–Panjang route, it is necessary to look at the costs incurred by vehicle operators using the route. As an illustration, vehicles that move from Java Island to Bandar Lampung City (Sumatra Island) have two alternatives, namely:

- Merak (Java Island)–Bakauheni (Sumatra Island) in Bandar Lampung City (via road = 85 km);
- Ciwandan (Java Island)–Panjang (Sumatra Island) in Bandar Lampung.

The primary difference in using the Merak–Bakuheni route is that it passes the highway, with the consequence of incurring more costs. Meanwhile, if one uses the Ciwandan–Pandang route, it is not necessary to pay for road transportation. If per day there are about 2000 vehicles that pass, with a cargo capacity of 45 tons per vehicle, then there will be 32.8 million tons per year that pass and potentially cause road damage. A comparison of the cost savings (efficiency) that can be obtained by vehicle operators can be seen in Table 5.

**Table 5.** Comparison of the cost efficiency of vehicle operators (truck owners).

| Cost | Merak–Bakauheni (IDR) | Ciwandan–Panjang (IDR) |
|---|---|---|
| Transport (crossing) cost | 1,113,500 | 1,123,000 |
| Fuel oil | 394,800 | |
| Oil and tire maintenance costs | 77,000 | |
| Total cost | 1,585,300 | 1,123,000 |
| Savings per truck (29.16%) | | 462,300 |
| With an estimated 2000 trucks per day, the savings obtained are: | | |
| Per day | | 924,600,000 |
| Per month | | 27,738,000,000 |
| Per year | | 332,856,000,000 |

The cost savings made by vehicle operators per unit vehicle (truck) when using the Ciwandan–Panjang SSS reaches 29.16%, or IDR 462,300. If it is assumed that there are around 2000 vehicles that pass per day, then the total savings can reach IDR 27.7 billion per month, and in a year, IDR 332.8 billion. This result is also in line with the research of M. M. Ramalho and T. A. Santos [5], who found that efficiency due to the application of SSS was 20–30% in Portugal. Likewise, SSS in Busan (Korea–Japan) experienced a decrease in total logistics costs [32].

### 3.6. Estimated Reduction in $CO_2$ Emissions

The number of vehicles operating on the Merak–Bakuheni route is proportional to the high exhaust emissions. The negative impact to the environment and health is very high. Body organs such as the lungs and blood vessels are hampered, and the eyes and skin are irritated. In addition, air pollution due to dust particles can cause chronic respiratory diseases, such as bronchitis and even lung cancer [35].

If every day an average of 5285 vehicles pass, then, based on IPCC research [34], it is estimated that around 122 kg of exhaust emissions will be generated. By diverting part of the route to SSS Ciwandan–Panjang, as many as 2000 vehicles, emissions can be reduced by around 37%. The simulation of calculating the reduction in exhaust emissions is analyzed based on the distance that may be reduced by the operation of the Ciwandan–Panjang SSS, which is 85 km per trip, or 170 km round trip. Complete simulation calculations can be seen in Table 6.

**Table 6.** Simulation of losses due to $CO_2$ exhaust emissions.

| Item | Formula | Unit | Total |
|---|---|---|---|
| Reduced distance (round trip) of 85 km | (a) | km | 170.00 |
| Number trucks/hour (2000/24) | (b) | trucks/h | 83.33 |
| IPCC $CO_2$ emission factor | (c) | Grams/L | 2924.90 |
| Specific energy consumption | (d) | L/km | 0.19 |
| Average $CO_2$ emissions in grams | (e) = (b) × (c) × (d) | g/h/km | 46,681.40 |
| Average $CO_2$ emissions in kilograms | (f) = ((e) × 1000) | kg/h/km | 46.68 |
| Average $CO_2$ emissions over a distance | (g) = (a) × (f) | kg/h | 7935.84 |
| Average $CO_2$ emissions in tonnes | (h) = (g)/1000 | tonnes/h | 7.94 |
| Average $CO_2$ emissions in years | (i) = (h) × 365 | tonnes/year | 241,381.76 |
| Pollutant cost (research Canada) | (j) | CAD | 205.00 |
| Kurs CAD | (k) | IDR | 10,326 |
| Loss | (l) = (i) × (j) × (k) | IDR | 511,063,117,224 |

Table 6 shows that based on an IPCC emission factor [34] of 2924.90 g/L and a specific energy consumption of 0.19 L/km, the average reduced emission is 46.68 kg/h/km, or around 7935.84 kg/h. If converted into tons/year, the average amount of emissions that can be reduced is 241,381.76 tons/year.

The very large losses that may be incurred if valued monetarily can be calculated using the amount of pollutants. The cost of pollutant losses when converted into monetary units reaches CAD 205 (exchange rate of IDR 10,326), so that the cost of losses incurred in a year reaches IDR 511 billion per year. The results of this study are also in line with Shamsi et al. [36] from Canadian urban roads, which obtain emission cost efficiencies of up to 6 million per year. As is the case in Beijing, reductions in energy consumption and pollution are also carried out through urban transport [37]. The research of M. Svindland and H. M. Hjelle [16], and D. Ulker et al. [15], also confirmed the advantages of container shipping with respect to $CO_2$ efficiency with SSS.

*3.7. The Operational Impact of the Ciwandan–Panjang SSS*

The operation of the Ciwandan–Panjang SSS has an impact on services on the Merak–Bakuheni route. The positive impacts include, among others, a diversion of the flow of vehicles by 36% (approximately 2000 vehicles per day). Waiting times for vehicles at Merak port will be reduced and queues will be reduced. The BOR of Merak and Bakauheni ports will decrease. The transfer of part of the fleet from Merak–Bakuheni to Ciwandan–Panjang will increase fleet utilization, which is currently only 34% per month. On the other hand, the performance of the wharf service (BOR) at the Ciwandan and Panjang ports will increase towards the optimal point.

During in-depth interviews, vehicle owners stated that they would benefit from the operation of the SSS, including reducing the risk of damage to goods, reducing transportation costs as a result of a reduction in vehicle operating costs, and increasing fuel efficiency. These results are in line with research by the Indonesian Trucking Association in 2018 [30]. The impacts obtained by vehicle drivers include reducing the risk of accidents due to fatigue, increasing driver endurance due to the driver having free time to rest on the boat, increasing mobility with adequate rest, and reducing mileage [38].

Another advantage for the general public and other road users is the creation of new jobs, reviving micro, small, and medium enterprises (MSMEs) in locations around ports,

reducing air pollution ($CO_2$ emissions), reducing congestion on highways, reducing road loads, reducing road maintenance costs, and reducing vehicle density [38].

## 4. Conclusions

### 4.1. Summary of Results

This paper assesses the probability of implementing SSS in terms of benefits received by stakeholders, cost efficiency by vehicle owners and operators, and the possibility of reducing $CO_2$ emissions from vehicle exhaust gases. The very high flow of vehicles between the islands of Java and Sumatra causes the problem of vehicles queuing on the Merak–Bakuheni ferry line. The performance of transportation services is impacted in the form of high transportation costs, long travel times, increasing $CO_2$ pollution, and road damage. Implementation of the Ciwandan–Panjang SSS can be an alternative to increase logistics flows between the two islands in order to increase the economy of the regions. The SSS of Ciwandan–Panjang is more competitive than the Merak–Bakuheni ferry. Nevertheless, Merak-Bakauheni and Ciwandan–Panjang will need each other and should be integrated. SSS operations can handle 32.8 million tons of cargo per year. The impact of the decrease in logistics costs is due to the efficiency of transportation costs, which reached 29.16%, or IDR 462,300 per vehicle (truck). The government will receive benefits in the form of NTSI and VAT revenues, while the IPC will receive income as a port operator. Assuming that there will be a shift of truck vehicles of around 2000 units per day, it is predicted that there will be a decrease in average vehicle exhaust emissions of 241,381.76 tons/year, or IDR 511 billion/year.

### 4.2. Limitations and Suggestions for Future Research

This research is limited to the study of benefits received by stakeholders and comparisons of cost efficiency. Furthermore, there are still gaps that can be investigated in the context of the operation of the Ciwandan–Panjang SSS, including: (1) the impact of the operation of the Ciwandan–Panjang SSS on the existence of the Merak–Bakauheni ferry; (2) the design and size of the ship in accordance with port facilities; (3) the feasibility of the economic and financial aspects of ship operators; (4) the technical feasibility of the port infrastructure to serve ships; (5) the types of vehicles to be loaded; and (6) ship scheduling.

**Author Contributions:** Conceptualization, D.A., E.M. and J.M.; methodology, D.A., E.M., W.P.H. and J.M.; validation, E.M., W.P.H. and F.S.P.; formal analysis, D.A., E.M., J.M., M. and T.S.M.; investigation, W.J.S., T.P. and A.K.; resources, D.A., E.M., T.P. and A.K.; data curation, W.P.H., F.S.P., M. and T.S.M.; writing—original draft preparation, D.A., J.M., W.P.H. and A.K.; writing—review and editing, M., W.J.S., T.P. and A.K.; visualization, F.S.P. and T.S.M.; supervision, J.M., M. and F.S.P.; project administration, W.J.S. and T.P. All authors have read and agreed to the published version of the manuscript.

**Funding:** This research received no external funding.

**Data Availability Statement:** The data presented in the current study are available upon reques from the corresponding author.

**Acknowledgments:** The authors would like to thank the Research and Development Center for Sea, River, Lake and Ferry Transportation, Ministry of Transportation, for their support in organizing the Focus Group Discussion forum.

**Conflicts of Interest:** The authors declare no conflict of interest.

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
