# Peer review of "Cost Efficiency and CO2 Emission Reduction in Short Sea Shipping: Evidence from Ciwandan Port–Panjang Port Routes, Indonesia"

_sustainability, doi:10.3390/su14106016_

Round 1

Reviewer 1 Report

This is a study on Cost Efficiency and CO2 Emission Reduction in Short Sea Shipping: Evidence From Ciwandan Port – Panjang Port Routes, Indonesia. The purpose of this research is to evaluate the possibility of opening the operation of the SSS route for the Ciwandan Port - Panjang Port, impact on reduce the density/overload of the road, reduce the number of /number of accidents, reduce congestion and reduce emissions. The topic is hot and several recent Ph.D. positions, scholarships, etc. are defined on this topic. The quality of the paper is good but some important concerns need to be considered:

  1. Abstract needs to be re-written, background, aim, method, results, and conclusion should be presented in order.
  2. The introduction needs reframing, in its current style, it is not consistent and informative. to do this task follow the stages of background: in which you should talk about the importance of the topic, literature: you should review the previous publications in this field, Research gaps: talk about the gaps in previous research and necessity of your own, aims: present your aims to fill the gaps of research, structure: mention the structure of your paper.
  3.  A flowchart is needed in the method section to clearly explain the stream of your methodology.
  4. Then a discussion section is needed and you should compare your finding with other sources and compare the methods with each other.
  5. The conclusion should be: A summary of the paper, highlights of your results and general conclusions from those findings, limitations of the research, and recommendations for future researchs.

Also i recommend to discuss the green taxes and overall trend of oil to sustainable economic transition using following references: 

https://www.sciencedirect.com/science/article/abs/pii/S1364032121012338

https://onlinelibrary.wiley.com/doi/10.1002/er.6762

https://so05.tci-thaijo.org/index.php/TER/article/view/257039

Author Response

Thank you for the suggestions on improving this paper.

We have made improvements to the feedback provided.

starting from the introduction, methodology, results, and discussion, and conclusions.

Reviewer 2 Report

Dear Authors,

You presented interesting data. In my opinion you can improve the overall quality of the manuscript if you will develop the presented results. You put a lot of tables, it could be, but I miss units and descriptions of the presented data. Of course, Indonesian people know what is Pelindo, but the overall merit of the article should be easy to understand for experts from other part of the World as well. I encourage you to use graphs as well to present your data, it will be more visuable. The title of your manuscript contains 'cost efficiency', in my opinion these results should be more highlighted.

Author Response

Thank you for the suggestions in improving this paper. we have made improvements to the feedback provided. starting from the introduction, methodology, results and discussion, and conclusions.

Round 2

Reviewer 2 Report

Dear Authors,

You did a very good work!